# Cutaneous and Subcutaneous Tumours of Small Pet Mammals—Retrospective Study of 256 Cases (2014–2021)

**DOI:** 10.3390/ani12080965

**Published:** 2022-04-08

**Authors:** Iwona Otrocka-Domagała, Katarzyna Paździor-Czapula, Joanna Fiedorowicz, Mateusz Mikiewicz, Agnieszka Piotrowska, Michał Gesek

**Affiliations:** Department of Pathological Anatomy, Faculty of Veterinary Medicine, University of Warmia and Mazury in Olsztyn, Oczapowskiego St. 13, 10-719 Olsztyn, Poland; i.otrocka-domagala@uwm.edu.pl (I.O.-D.); fiedorowicz.joanna@gmail.com (J.F.); mateusz.mikiewicz@uwm.edu.pl (M.M.); cytologika@gmail.com (A.P.); michal.gesek@uwm.edu.pl (M.G.)

**Keywords:** oncology, sarcoma, rodents, immunohistochemistry, histopathology, tumour

## Abstract

**Simple Summary:**

Several species of small mammals are very popular as companion pet animals and therefore demand professional veterinary care, including proper diagnostic and treatment procedures. The incidence of neoplastic diseases in companion pet animals has increased over time, as it has in humans. The aim of this study was to evaluate the incidence of cutaneous tumours in small mammal pets, including guinea pigs, rats, pet rabbits, ferrets, hamsters, degus, African pygmy hedgehogs, Mongolian gerbils and chinchillas, submitted for histopathology in 2014–2021. Malignant tumours represented the predominant group of cutaneous tumours in rats, African pygmy hedgehogs, degus and chinchillas, and represented a significant percentage of lesions in guinea pigs, pet rabbits, ferrets, hamsters and Mongolian gerbils. We also described the first case of melanocytoma in a pet rabbit, epitheliotropic T-cell lymphoma in a degu, cutaneous histiocytic sarcoma in a Mongolian gerbil, fibrosarcoma in two chinchillas and epithelioid haemangioma in a chinchilla. The incidence of malignant neoplasms among spontaneous integumentary tumours submitted for histopathology is high in many species of small mammal pets. Therefore, each cutaneous tumour should be sampled for further diagnosis and treatment.

**Abstract:**

Since small mammals are gaining popularity as pets in Poland, the number of tumour samples submitted for histopathological examination is quite high. This study was a retrospective analysis of cutaneous and subcutaneous tumours in small pet mammals submitted for histopathology in 2014–2021. The analysis included 256 tumours sampled from 103 guinea pigs, 53 rats, 43 pet rabbits, 21 ferrets, 17 hamsters, 8 degus, 5 African pygmy hedgehogs, 3 Mongolian gerbils and 3 chinchillas. Tumours were diagnosed based on routine histopathology, with additional immunohistochemistry when necessary. The results of this study revealed that the vast majority of cutaneous tumours in guinea pigs were benign, with a predominance of lipoma. Adnexal tumours constituted a significant percentage of cutaneous tumours in guinea pigs (24.3%, with the most common being trichofolliculoma), pet rabbits (46.5%, with the most common being trichoblastoma), ferrets (33.3%, mostly derived from sebaceous glands), hamsters (52.9%, with the most common being trichoepithelioma) and gerbils (66.7%, scent gland epithelioma). Soft tissue sarcomas were a predominant group of tumours in rats (52.8%, with the most common being fibrosarcoma), African pygmy hedgehogs (100%), degus (87.5%) and chinchillas (66.7%). Melanocytic tumours were only sporadically seen in small mammal pets. Mast cell tumours were diagnosed only in ferrets, while epitheliotropic T-cell lymphoma was diagnosed only in a hamster and a degu. In summary, malignant tumours constitute a significant percentage of cutaneous tumours in many species of small mammal pets. Therefore, each cutaneous tumour should be sampled for further cytologic or histopathologic diagnosis.

## 1. Introduction

Small mammals are commonly used in laboratory research and are also gaining popularity as companion pet animals. In recent years, we have also observed an increase in the incidence of spontaneous neoplastic diseases in these animals; thus, attention has been given to the need to develop diagnostic, therapeutic and prophylactic methods in small mammal oncology. To develop and identify prognostic and predictive factors for oncologic therapy, it is necessary to understand the nature and characteristics of neoplastic lesions in small mammals. In addition, small mammals, like other domestic and wild animals, are exposed to the same environmental carcinogenic risk factors as humans and can be used as sentinels for environmental and public health [1].

Therefore, knowledge regarding tumours in small domestic mammals develops dynamically, and recent studies have revealed that cutaneous and subcutaneous tumours constitute a significant percentage of all tumours diagnosed in guinea pigs, gerbils [2], pet rabbits [3], African pygmy hedgehogs [4] and ferrets [5]. However, despite numerous studies, data on the incidence of particular cutaneous and subcutaneous tumour types in small mammal pets have varied and appear to change over time [2,3,4,6,7,8]. Reports of cutaneous and subcutaneous tumours in pet rats, gerbils, hamsters, degus and chinchillas have been rather sparse; therefore, it is important to update these data, which could be essential for the appropriate clinical management and oncological care of these small mammals.

The aim of this retrospective study was to analyse the incidence of cutaneous and subcutaneous tumours in various species of small pet mammals, including guinea pigs, rats, pet rabbits, ferrets, hamsters, degus, African pygmy hedgehogs, Mongolian gerbils and chinchillas, submitted for histopathology to the Department of Pathological Anatomy, Faculty of Veterinary Medicine, University of Warmia and Mazury in Olsztyn, Poland. Moreover, the assessment of the incidence of particular types of neoplasms depending on the sex and age of the animals was performed.

## 2. Materials and Methods

The retrospective analysis included 256 cutaneous tumour samples obtained by either excisional or incisional biopsy from 103 guinea pigs (*Cavia porcellus*), 53 rats (*Rattus norvegicus domestica*), 43 rabbits (*Oryctolagus cuniculus domesticus)*, 21 ferrets (*Mustela putorius furo*), 17 hamsters (7 Djungarian—*Phodopus sungorus*, 7 golden—*Mesocricetus auratus*, 1 Ciscaucasian—*Mesocricetus raddei* and 2 without species specification), 8 degus (*Octodon degus*), 5 African pygmy hedgehogs (*Atelerix albiventris*), 3 Mongolian gerbils (*Meriones unguiculatus*) and 3 chinchillas (*Chinchilla lanigera*) during 2014–2021 (stored in the archive of the Department of Pathological Anatomy, Faculty of Veterinary Medicine, University of Warmia and Mazury in Olsztyn). The mammary tumours were excluded from the study. The tissue samples were fixed in 10% buffered formalin, embedded in paraffin and cut into 3 μm sections. The sections were processed routinely and stained with Mayer’s haematoxylin and eosin (HE). In poorly differentiated sarcomas and lymphomas, additional immunohistochemistry was applied. Antigen retrieval was conducted in a PT-Link (Dako, Glostrup, Denmark) using Tris-EDTA buffer with pH = 9. Immunohistochemical examination was performed manually using appropriate primary antibodies, collected in Table 1, and a visualisation system based on the immunoperoxidase method with 3.3-diaminobenzidine (DAB) as a substrate. The slides were counterstained with Mayer’s haematoxylin. Positive and negative control slides were processed together with the evaluated sections. The reactivity of antibodies in small mammal tissues is summarised in Table 2.

## 3. Results

### 3.1. Guinea Pigs

The mean age of affected guinea pigs was 3.5 years old (age range: 8 months–9 years), with 54 females, 47 males and two with sex unspecified. The soft tissue tumours constituted the vast majority of cutaneous tumours in guinea pigs (76 cases) and were derived mostly from adipocytes or fibroblasts. The most frequently encountered tumour was lipoma (46 cases), including 40 cases of simple lipoma (26—solitary; 14—multiple), four cases of fibrolipoma and two cases of infiltrative lipoma. The most common location of lipomas was the inguinal area (16/26 cases of simple lipoma), the other localisations included trunk, neck, axilla and head. The other soft tissue tumours included 13 cases of fibrosarcoma, nine cases of liposarcoma (eight—solitary; one—multiple), three cases of fibroma, two cases of haemangioma (capillary and mixed capillary–cavernous type) and a single case each of the following: haemangiosarcoma, perivascular wall tumour and poorly differentiated sarcoma. In the poorly differentiated sarcoma, tumour cells expressed vimentin (clone 3B4) but were negative for Iba-1, α-SMA and desmin. After lipomas, the second most frequent group of tumours was benign adnexal tumours (23 cases), with a prevalence of trichofolliculoma (15 cases) followed by trichoepithelioma (seven cases, six solitary and one multiple) and a single case of sebaceous adenoma. There were also single cases of epitheliomatous sebaceous carcinoma, apocrine adenocarcinoma, squamous cell carcinoma and compound melanocytoma.

### 3.2. Rats

The mean age of affected rats was 21.9 months old (age range: 6–36 months), with 38 males, 14 females and one with sex unspecified. Soft tissue sarcomas constituted the vast majority of analysed tumours (28 cases), represented by fibrosarcoma (15 cases), poorly differentiated sarcoma (five cases), perivascular wall tumour (four cases), myxosarcoma (two cases) and malignant peripheral nerve sheath tumour (two cases). In two poorly differentiated sarcomas, tumour cells expressed Iba-1 and vimentin (clone V9) and were negative for desmin and α-SMA, so they were reclassified as histiocytic sarcomas (Figure 1). In the three remaining poorly differentiated sarcomas, tumour cells expressed vimentin (clone V9) and were negative for Iba-1, desmin and α-SMA, so they were classified as pleomorphic fibrosarcomas, according to Greaves et al. [9]. Benign soft tissue tumours were less frequent and represented by fibroma (nine cases, eight solitary and one multiple), simple lipoma (two cases) and single cases of benign peripheral nerve sheath tumour and neurofibroma. Squamous cell carcinoma (eight cases) constituted the majority of epithelial tumours, while benign adnexal tumours were represented by single cases of sebaceous adenoma, apocrine ductal adenoma (solid-cystic type) and pilomatricoma. There was also a single case of melanocytoma.

### 3.3. Rabbits

The mean age of the affected rabbits was 6.1 years old (age range: 1–12 years): 27 were males, 14 were females, and two were unspecified. The most common cutaneous tumour in pet rabbits was trichoblastoma trabecular type (18 cases, 17 solitary, one multiple), followed by fibrosarcoma (seven cases). Other epithelial tumours included two cases of squamous cell carcinoma and one case each of the following: apocrine adenocarcinoma, basal cell carcinoma, trichoepithelioma and papilloma. Other soft tissue tumours included three cases of lipoma (two simple lipomas and one fibrolipoma), three cases of fibroma and one case each of the following: benign peripheral nerve sheath tumour, malignant peripheral nerve sheath tumour and haemangioma (inflammatory lobular capillary type). Melanocytic tumours were represented by two cases of melanoma and one case of compound melanocytoma (Figure 2).

### 3.4. Ferrets

The mean age of the affected ferrets was 4.6 years old (age range: 1.5–8 years; males: 15; females: five; unspecified: one). The most common cutaneous tumour in ferrets was mast cell tumour (seven cases), followed by squamous cell carcinoma (four cases) and sebaceous epithelioma (four cases). In one of the ferrets with mast cell tumours, a new tumour developed several months after the first surgery. Other tumour types included two cases of poorly differentiated sarcomas (localised in the inguinal area and at the base of the tail) and single cases of sebaceous carcinoma, secretory apocrine adenoma, apocrine adenocarcinoma and malignant peripheral nerve sheath tumour. In poorly differentiated sarcomas, tumour cells expressed vimentin (clone V9) and α-SMA and were negative for Iba-1 and desmin (Figure 3).

### 3.5. Hamsters

The mean age of affected hamsters was 14.2 months old (age range: 7–24 months), with 10 males and seven females. All epithelial tumours were of adnexal origin and included trichoepithelioma (five cases, four solitary, one multiple) and one case each of the following: trichofolliculoma, sebaceous epithelioma, apocrine adenocarcinoma and apocrine ductular carcinoma. Trichoepithelioma was diagnosed in two golden hamsters, one Djungarian hamster, and one Ciscaucasian hamster, and in one case, the species was unspecified. Other adnexal tumours were diagnosed in Djungarian hamsters. Mesenchymal tumours (six cases) included three cases of fibrosarcoma (diagnosed in one golden hamster, one Djungarian hamster and one unspecified species); one case of poorly differentiated sarcoma (tumour cells expressed vimentin, clone V9 and α-SMA and were negative for desmin and Iba-1), diagnosed in a Djungarian hamster; one case of simple lipoma, diagnosed in a golden hamster; and one case of epitheliotropic T-cell lymphoma (tumour cells expressed CD3+ and were CD79a and CD20 negative; Figure 4), diagnosed in a golden hamster. There were also two cases of melanoma (both in golden hamsters).

### 3.6. Other Species

In degus (mean age: 4.2 years old, age range: 1.5–6 years; females: four; males: three; sex unspecified: one), except for one case of epitheliotropic T-cell lymphoma (CD3+), all tumours were soft tissue sarcomas: fibrosarcoma—two cases; poorly differentiated sarcoma (vimentin, clone V9+, Iba-1-, desmin-, α-SMA-)—two cases; myxosarcoma—one case; anaplastic sarcoma with giant cells (vimentin, clone V9+, Iba-1-, desmin-, α-SMA-; Figure 5)—one case; and malignant peripheral nerve sheath tumour—one case. In African pygmy hedgehogs (mean age 4.4 years old, age range: 2–7 years; three males, two females), all cutaneous tumours were represented by soft tissue sarcomas: fibrosarcoma, liposarcoma, malignant peripheral nerve sheath tumour, histiocytic sarcoma (tumour cells expressed Iba-1+ and vimentin, clone 3B4, and were negative for desmin and α-SMA-; Figure 6) and poorly differentiated sarcoma (vimentin clone 3B4+, Iba-1-, desmin-, α-SMA-). In Mongolian gerbils, there were two cases of abdominal scent gland epithelioma (both two-year-old males) and one case of histiocytic sarcoma located in the nasal area (age unknown, female; tumour cells expressed Iba-1 and vimentin, clone V9, and were negative for desmin and α-SMA-). In chinchillas, there were two cases of fibrosarcoma (14-year-old female and 17-year-old male, Figure 7) and one case of an epithelioid haemangioma (age unknown, male, Figure 8).

Detailed information about the incidence of particular tumour types in small mammal pets, depending on sex and age, is summarised in Table 3.

## 4. Discussion

In the present study involving numerous small mammal species, guinea pigs were over-represented, reflecting the popularity of these animals as pets [10]. Cutaneous tumours were diagnosed mostly in middle-aged guinea pigs (3–4 years old), and the average age was similar in different tumour types, except lipomas, which seemed to involve slightly younger animals (mean age: 2.9 years old), similar to a previous study [11]. In contrast, Minarikova reported that cutaneous tumours were significantly more often diagnosed in guinea pigs younger than 2 years of age [7]. The vast majority of cutaneous tumours observed in guinea pigs were benign, consistent with previous reports [7,11]. Previously, it was shown that the most common cutaneous tumour in guinea pigs was trichofolliculoma, a benign neoplasm characterised by abortive follicular adnexal structures in the walls of cysts lined by squamous epithelium [2,11,12], sometimes misinterpreted as trichoepithelioma [2,13]. However, in our study, the most frequently diagnosed benign tumour was lipoma. According to previous reports, lipoma is common in guinea pigs and occurs as a single mass or multiple tumours [11,13], as also seen in the present study. Furthermore, we observed a predilection for the inguinal area, as also noted by Kanfer and Reavil [11]. In the present study, trichofolliculoma was the second most common cutaneous tumour in guinea pigs and was diagnosed far more frequently in females, whereas previous studies have suggested a predilection in males [13,14]. Although the majority of cutaneous tumours in guinea pigs were benign, we showed for the first time that a significant percentage constituted malignant tumours, represented mostly by soft tissue sarcomas (24%), with a predominance of fibrosarcoma followed by liposarcoma. In previous studies, soft tissue sarcomas have been only sporadically reported in guinea pigs [7,11,13]. However, Hawkins and Bishop noted that liposarcoma had a relatively high prevalence in the species [14]. While fibrosarcoma is quite common in domestic animals [15], liposarcoma is rare [15,16].

Most reports of spontaneous tumours in rats are related to laboratory animals [12]. Although the overall incidence of tumours in rats is high, cutaneous and subcutaneous tumours are relatively infrequent [2]. However, among all dermatologic conditions in pet rats, cutaneous tumours were most commonly diagnosed in one of the veterinary teaching hospitals [17]. In the present study, companion rats constituted the second most numerous group of small mammal pets diagnosed with skin tumours, suggesting that these tumours are not uncommon. Cutaneous and subcutaneous tumours were diagnosed mostly in middle-aged to older rats, with a mean age of 21.9 months old. It was previously shown that the tumour incidence is especially high in rats older than two years old [18]. Interestingly, we observed a strong prevalence in males (73%), which has never before been shown. In contrast to guinea pigs, the majority of diagnosed cutaneous and subcutaneous tumours in rats were malignant. We showed that the most common tumours were soft tissue sarcomas, with a prevalence of fibrosarcoma. The results of previous studies have been contradictory. In one study, subcutaneous mesenchymal tumours in rats were more common than epithelial tumours [2], while in another study, squamous cell carcinoma was most common [17]. It was also previously shown that fibrosarcoma is common in middle-aged to older rats [13], and we obtained similar results. In our study, all cases of poorly differentiated sarcomas in rats underwent basic immunophenotyping. In two of them, tumour cells expressed the histiocytic marker Iba-1 and were classified as histiocytic sarcomas. Histiocytic sarcoma occurs quite frequently in various strains of laboratory rats, and the subcutis is the predilection site in Wistar rats [19,20]. The incidence of histiocytic sarcoma in companion rats is not known. We showed that Iba-1 could be a useful marker of histiocytic sarcoma in rats and could be applied in routine diagnostics for poorly differentiated sarcomas to classify these tumours more precisely. Previously, immunohistochemical markers used for histiocytic sarcomas in rats included vimentin, CD68 and lysozyme [21]. However, Iba-1 was used previously as a marker of macrophage/microglial origin of brain tumours in rats [22]. In three cases of poorly differentiated sarcomas, tumour cells expressed only vimentin and were classified as pleomorphic fibrosarcomas. Pleomorphic fibrosarcoma, previously known as malignant fibrous histiocytoma, is considered to constitute a group of undifferentiated primitive sarcomas diagnosed in laboratory animals and humans. In humans, these tumours are further subcategorised according to immunophenotype and ultrastructure [9]. We showed that pleomorphic fibrosarcomas constitute a significant percentage (10.7%) of soft tissue sarcomas in pet rats.

In this study, cutaneous tumours were seen mainly in middle-aged to older pet rabbits, and the majority of these tumours were benign, with the most frequent being trichoblastoma (trabecular type), consistent with previous studies [3,11,23,24]. Furthermore, the majority of cutaneous tumours were diagnosed in males. According to previous studies, male predilection was observed mostly or exclusively in mesenchymal tumours [23,24] and can result from the overall greater longevity of male rabbits [24]. Some mesenchymal neoplasms, such as fibromas (also known as collagenous hamartomas), occur exclusively in males, so hormonal pathogenesis is suspected [11,23,24]. In the current study, malignant tumours also constituted a significant percentage of all cutaneous tumours (32.6%) and included soft tissue sarcomas, melanomas and malignant tumours derived from the epidermis or adnexa. It was previously shown that the majority of cutaneous mesenchymal neoplasms in rabbits were malignant [24], but we observed that the percentages of benign and malignant mesenchymal tumours was equal. Melanomas were previously described in pet rabbits [23]. Although most of the previously reported melanomas were diagnosed in males [11,24], both cases of melanomas included in the current study were confirmed in females. Furthermore, we found one case of melanocytoma that has not been reported before in pet rabbits.

Tumours of the skin and subcutis are the third most common neoplasms in ferrets and are overtaken by islet cell tumours (insulinoma) and adrenocortical neoplasms [5]. In our study, as in recent studies [5,11,25], mast cell tumours constituted the most common skin neoplasm diagnosed in ferrets. Cutaneous mastocytoma in ferrets, unlike in dogs, is a universally benign neoplasm with good prognosis that does not spread locally or metastasise; therefore, surgical excision with good margins is curative [5,11]. We observed mostly isolated forms of this neoplasm, but mast cell tumours in ferrets can also have the characteristics of multiple concurrent lesions [26]. Mast cell tumours were more common in males, consistent with other results [11]; however, in our study, the mean age of the affected animals was 3 years old, while other researchers have indicated an average age of 4.5 to 5 years old for this tumour type [11,26]. Since the age range for mast cell tumours in ferrets is quite broad, ranging from two to nine years old, it seems that the low average age of mast cell neoplasm cases in our study might have resulted from the small number of diagnosed cases. However, it cannot be excluded that the phenomenon is caused by husbandry issues, unfavourable environmental factors or genetic predispositions. Sebaceous epithelioma and squamous cell carcinoma were the second ex aequo cutaneous neoplasms diagnosed in ferrets in our study. Sebaceous epitheliomas arise from pluripotential basal cells of sebaceous glands, are benign and slow growing and do not metastasise but can have locally aggressive potential; their complete excision is usually curative [5,11]. In our study, we did not find a sex predilection for this neoplasm, consistent with the results of other studies [11]; however, Parker and Picut [27] observed sebaceous epithelioma more often in females. It was previously shown that this tumour occurred in ferrets at an average age of 5.2 years old [11]; however, we diagnosed sebaceous epithelioma in slightly older individuals with an average age of 6.8 years old. In very rare cases, sebaceous epithelioma can give rise to squamous cell carcinoma [5,25]. Although we diagnosed four cases of squamous cell carcinoma, we found no evidence of its origin in or association with sebaceous glands. All our cases of squamous cell carcinoma were diagnosed in middle-aged males (mean age 3.8 years old), which might suggest a strong sex predilection; however, other studies have not confirmed this correlation [11]. Conversely, there have been single reports of multicentric squamous cell carcinoma in males [28,29]; in one of these, a relationship between tumour and papillomavirus infection was confirmed [29]. In our study, eosinophilic intranuclear inclusion bodies were not found in neoplastic keratinocytes, and additional tests were not performed. The age range of squamous cell carcinoma in ferrets is quite broad, between 3 and 6 years old (average age 4.5) [11], close to the results of our research. We also diagnosed two cases of poorly differentiated sarcoma expressing vimentin and α-SMA. Although none of these tumours were located in a routine vaccination site (groin and tail), their immunophenotype, in line with the results of other studies [30], could indicate that they were injection related; however, we did not have a vaccination history for these ferrets.

Integumental neoplasms constitute the majority of spontaneous tumours in Djungarian hamsters, while the incidence of these tumours in golden hamsters is low [31]. However, in the current study, these species were equally represented. The majority of integumental tumours in hamsters are of epithelial origin [32]. We observed that the most common cutaneous tumour was trichoepithelioma, followed by fibrosarcoma and melanoma. Except for trichoepithelioma, all adnexal tumours were diagnosed in Djungarian hamsters. It was previously shown that trichoepitheliomas can develop in golden hamsters due to hamster polyomavirus infection [33]. In our study, sarcomas constituted a significant percentage of cutaneous tumours in hamsters and included fibrosarcoma and a poorly differentiated sarcoma. In the poorly differentiated sarcoma, tumour cells expressed vimentin and α-SMA, suggesting possible myofibroblastic differentiation. Interestingly, we did not observe atypical fibroma derived from ganglion-cell-like cells, which is commonly diagnosed in Djungarian hamsters, or its malignant counterpart [31,34]. Cutaneous melanomas were previously reported in golden hamsters but with a strong male predisposition [12]. However, in the present study, both cases of melanoma were observed in females, although the overall number of golden hamsters in the current study is too small to draw any conclusions about sex predispositions. We also observed one case of epitheliotropic T-cell lymphoma, which was previously reported in golden hamsters [35].

The incidence of neoplasia in degus is low, and tumours are sporadically seen in older animals [36]. In the present study, almost all cutaneous tumours observed in degus were soft tissue sarcomas, and the age range was 1.5–6 years old, suggesting that these tumours occur not only in older animals. It was previously reported that the most common tumour in degus was cutaneous fibrosarcoma [37]. We also observed one case of epitheliotropic T-cell lymphoma, which has not been reported before in degus.

It was previously shown that the majority of tumours in African pygmy hedgehogs were malignant, with integument as a common location [4,8,38]. In the present study, all tumours diagnosed in African pygmy hedgehogs were malignant and represented by various types of soft tissue sarcomas. A previous study reported that most cutaneous tumours in hedgehogs were of mesenchymal origin, with the most common being fibrosarcoma [8]. Interestingly, we reported one case of histiocytic sarcoma confirmed by immunoexpression of Iba-1. Histiocytic sarcoma was previously described in African pygmy hedgehogs and occurs in localised (cutaneous) and disseminated (visceral) forms [39].

Although the incidence of spontaneous tumours in Mongolian gerbils is quite high [40], the number of cutaneous and subcutaneous tumours included in the present study was low (three cases). These rodents may not be as popular as companion animals as other small mammal species included in this study, or presumably not all excised cutaneous tumours are sent for histopathology. We reported two cases of abdominal scent gland epitheliomas diagnosed in two-year-old males. Scent gland epitheliomas have been previously diagnosed in males at a similar age [41]. We also reported one case of histiocytic sarcoma, and the histiocytic origin of tumour cells was confirmed by immunoexpression of Iba-1. Disseminated histiocytic sarcoma involving internal organs and bone marrow was previously diagnosed in a 59-month-old female Mongolian gerbil; however, the phenotype of neoplastic cells was not confirmed by immunohistochemistry [42]. To the best of our knowledge, our case is the first cutaneous histiocytic sarcoma confirmed immunohistochemically in a Mongolian gerbil.

There is a paucity of information in the literature regarding cutaneous and subcutaneous tumours in chinchillas [2,13]. We diagnosed two cases of fibrosarcoma in a female and a male, both of advanced age, and one epithelioid haemangioma in a male. Neither type of neoplasm has been previously described in the skin and subcutis of chinchillas.

## 5. Conclusions

In conclusion, cutaneous and subcutaneous tumours constitute a large and heterogeneous group of neoplasms in small pet mammals, with a significant percentage of malignancies among all cutaneous tumour samples submitted for histopathology in Poland. The incidence of soft tissue sarcomas is high in guinea pigs and rats; these tumours prevail in degus, African pygmy hedgehogs and chinchillas. The high incidence of malignancies observed in this study could be the result of environmental and food pollution or other breeding or behavioural factors. However, this study analysed retrospectively histopathological submissions, and possibly malignant tumours could be more likely to be sampled for histopathology than grossly benign nodules. Furthermore, we observed the first case of melanocytoma in a pet rabbit, epitheliotropic T-cell lymphoma in a degu, cutaneous histiocytic sarcoma in a Mongolian gerbil, fibrosarcoma in two chinchillas and epithelioid haemangioma in a chinchilla. The results of our study indicate that Iba-1 could be a useful marker of histiocytic sarcoma in small pet mammals and should be applied in routine diagnostics for poorly differentiated sarcomas. It is essential to monitor neoplastic diseases in small pet mammals to improve diagnosis and treatment.

## Figures and Tables

**Figure 1 animals-12-00965-f001:**
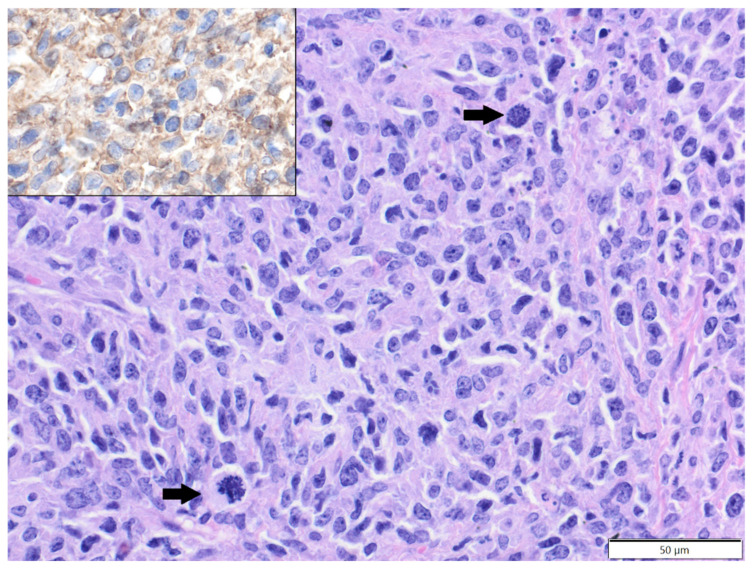
Histiocytic sarcoma, submandibular area, rat. Pleomorphic tumour cells are arranged haphazardly and show mitotic figures (arrows). HE. Inset: tumour cells express Iba-1. IHC.

**Figure 2 animals-12-00965-f002:**
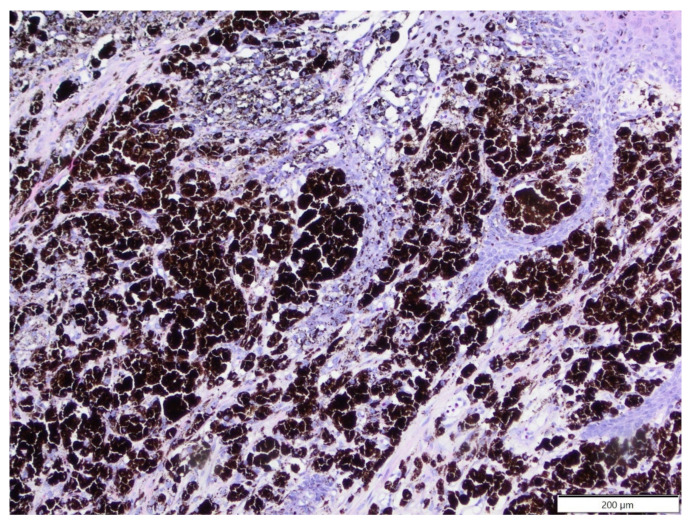
Compound melanocytoma, facial area, pet rabbit. Heavily melanised, well-differentiated tumour cells formed solid sheets and massively infiltrated the epidermis. HE.

**Figure 3 animals-12-00965-f003:**
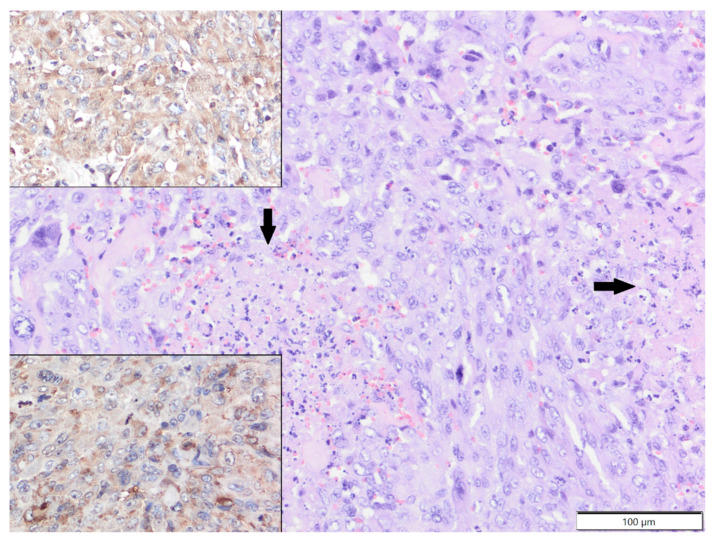
Poorly differentiated sarcoma, tail, ferret. Tumour cells show high anisocytosis and anisokaryosis and undergo necrosis (arrows). HE. Upper inset: tumour cells show cytoplasmic expression of vimentin. IHC. Lower inset: tumour cells show slight cytoplasmic expression of α-SMA. IHC.

**Figure 4 animals-12-00965-f004:**
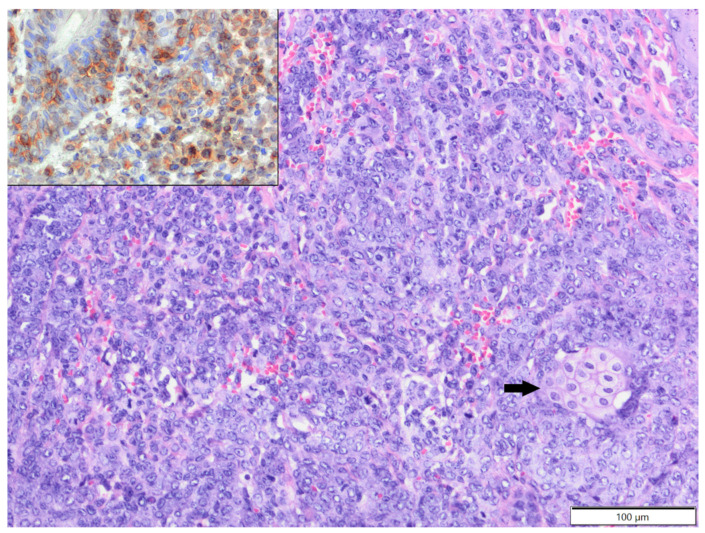
T-cell epitheliotropic lymphoma, tail, golden hamster. Large lymphocytes massively infiltrated the skin and adnexa, and there is a single entrapped sebaceous gland visible (arrow). HE. Inset: tumour cells show expression of CD3. IHC.

**Figure 5 animals-12-00965-f005:**
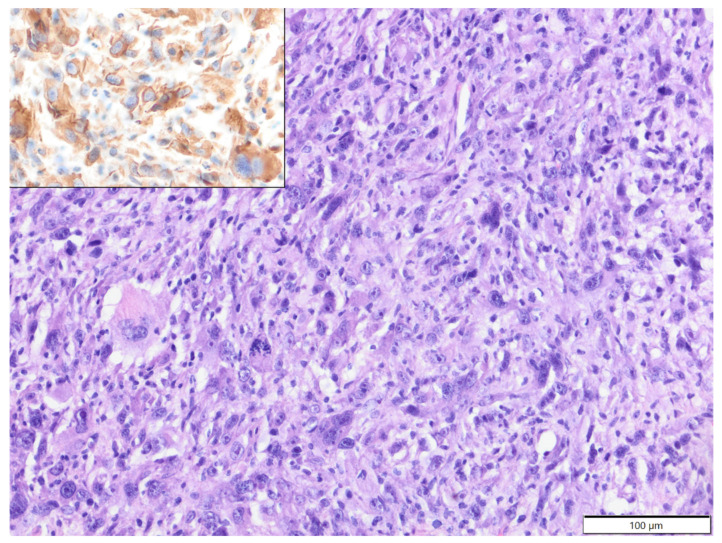
Anaplastic sarcoma with giant cells, neck, degu. The tumour cells show high levels of anaplasia and multinucleation. HE. Inset: tumour cells show cytoplasmic expression of vimentin. IHC.

**Figure 6 animals-12-00965-f006:**
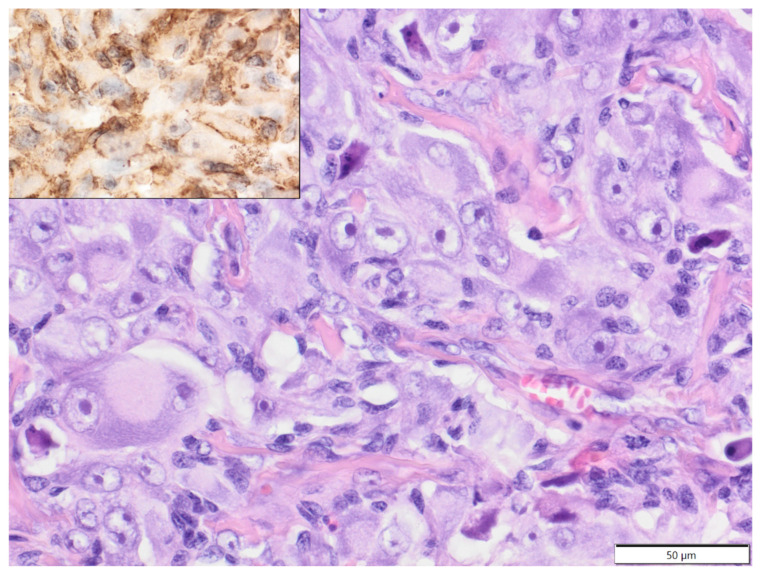
Histiocytic sarcoma, forelimb, African pygmy hedgehog. Pleomorphic tumour cells show high anisocytosis and anisokaryosis. Nuclei are large with marginated chromatin and distinct, single nucleoli. HE. Inset: tumour cells show expression of Iba-1. IHC.

**Figure 7 animals-12-00965-f007:**
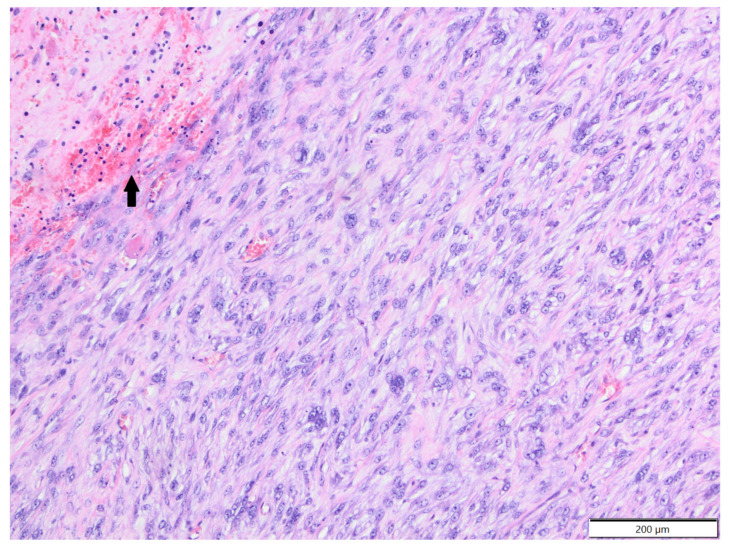
Fibrosarcoma, area unspecified, chinchilla. There are bundles of spindle to oval tumour cells and moderate amount of collagen fibres between them. Tumour cells undergo focal necrosis (arrow). HE.

**Figure 8 animals-12-00965-f008:**
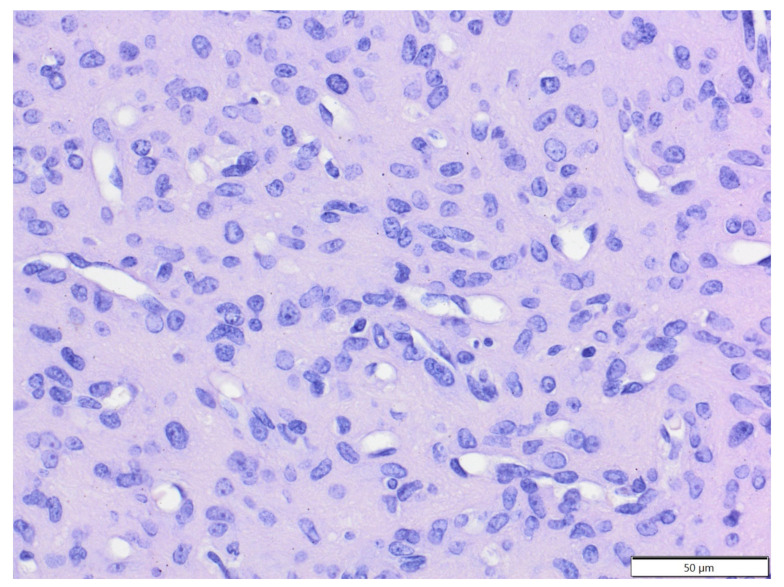
Epithelioid haemangioma, facial (nasal) area, chinchilla. Large and plump tumour cells form small blood vessels and solid aggregates. HE.

**Table 1 animals-12-00965-t001:** Primary antibodies and visualisation systems used in immunohistochemical examination.

Primary Antibody	Clone	Dilution	Source	Visualisation System
Iba-1	Polyclonal rabbit	1:200	Wako Pure Chemical Industries, Ltd., Osaka, Japan	Impress HRP Universal Antibody (Anti-Mouse/Rabbit IgG) ^a^
CD3	Polyclonal rabbit anti-human	1:50	Dako, Glostrup, Denmark	Impress HRP Universal Antibody (Anti-Mouse/Rabbit IgG) ^a^
CD79a	Monoclonal mouse anti-human HM57	1:100	Bio-Rad Laboratories Inc., Hercules, CA, USA	EnVision+ System-HRP, Mouse (DAB) ^b^
CD20	Monoclonal rabbit anti-human SP32	1:100	Abcam, Cambridge, UK	Impress HRP Universal Antibody (Anti-Mouse/Rabbit IgG) ^a^
Vimentin	Monoclonal mouse anti-bovine VIM 3B4	1:100	Dako, Glostrup, Denmark	EnVision+ System-HRP, Mouse (DAB) ^b^
	Monoclonal mouse anti-porcine V9	1:100		
Desmin	Monoclonal mouse anti-human D33	1:50	Dako, Glostrup, Denmark	EnVision+ System-HRP, Mouse (DAB) ^b^
α-SMA	Monoclonal mouse anti-human 1A4	1:50	Dako, Glostrup, Denmark	EnVision+ System-HRP, Mouse (DAB) ^b^

^a^ Vector Laboratories Inc., Burlingame, CA, USA. ^b^ Dako, Glostrup, Denmark.

**Table 2 animals-12-00965-t002:** Immunoreactivity of primary antibodies in control tissues of small mammal pets (“+” indicates that there was a positive reaction in control tissues; “-” indicates that there was lack of any reaction in control tissues; and “nd” indicates that the investigation was not performed due to either a lack of the appropriate control tissue or a lack of the appropriate primary antibody).

Species	Vimentin (V9)	Vimentin (3B4)	α-SMA	Desmin	Iba-1	CD3	CD79a	CD20
guinea pig	+	+	+	+	+	+	+	-
rat	+	-	+	+	+	+	+	+
pet rabbit	+	+	+	+	nd	nd	+	nd
ferret	+	-	+	+	+	+	+	+
golden hamster	+	-	+	+	+	+	+	+
Djungarian hamster	+	-	+	+	+	+	+	+
degu	+	+	+	+	+	+	-	+
African pygmy hedgehog	-	+	+	+	+	+	+	-
Mongolian gerbil	+	-	+	+	nd	nd	nd	nd
chinchilla	+	-	+	+	+	+	+	-

**Table 3 animals-12-00965-t003:** Incidence of particular tumour types in small mammal pets, depending on the sex and age.

Nr. of Cases	Tumour Type	Frequency	Females	Males	Mean Age
Guinea pigs					
46	Lipomas	44.7%	22	23	2.9 years
15	Trichofolliculoma	14.6%	11	4	3.9 years
13	Fibrosarcoma	12.6%	6	7	3.7 years
9	Liposarcoma	8.7%	4	4	3.6 years
7	Trichoepithelioma	6.8%	5	2	3.4 years
3	Fibroma	2.9%	2	1	3.8 years
2	Haemangioma	1.9%	0	2	2.5 years
1	Haemangiosarcoma	1%	0	1	9 years
1	Perivascular wall tumour	1%	0	1	2.5 years
1	Poorly differentiated sarcoma	1%	0	1	5.4 years
1	Sebaceous adenoma	1%	1	0	3 years
1	Epitheliomatous sebaceous carcinoma	1%	0	1	5 years
1	Apocrine adenocarcinoma	1%	0	1	5 years
1	Squamous cell carcinoma	1%	1	0	4 years
1	Compound melanocytoma	1%	0	1	5 years
Rats					
15	Fibrosarcoma	28.3%	4	11	17.9 months
9	Fibroma	17%	2	7	23.4 months
8	Squamous cell carcinoma	15.1%	1	7	24.3 months
4	Perivascular wall tumour	7.5%	1	3	21.3 months
3	Pleomorphic fibrosarcoma	5.7%	1	1	24 months
2	Histiocytic sarcoma	3.8%	1	1	20 months
2	Myxosarcoma	3.8%	0	2	31 months
2	Malignant peripheral nerve sheath tumour	3.8%	1	1	26 months
2	Simple lipoma	3.8%	1	1	27 months
1	Benign peripheral nerve sheath tumour	1.9%	0	1	24 months
1	Neurofibroma	1.9%	0	1	6 months
1	Sebaceous adenoma	1.9%	0	1	24 months
1	Apocrine ductal adenoma	1.9%	unspecified		18 months
1	Pilomatricoma	1.9%	1	0	24 months
1	Melanocytoma	1.9%	1	0	33 months
Pet rabbits					
18	Trichoblastoma	41.9%	5	11	6 years
7	Fibrosarcoma	16.3%	3	4	7.4 years
3	Lipomas	7%	3	0	2.3 years
3	Fibroma	7%	0	3	10.5 years
2	Melanoma	4.7%	2	0	7 years
2	Squamous cell carcinoma	4.7%	0	2	5.8 years
1	Apocrine adenocarcinoma	2.3%	0	1	6 years
1	Basal cell carcinoma	2.3%	1	0	5 years
1	Trichoepithelioma	2.3%	0	1	8 years
1	Papilloma	2.3%	0	1	3 years
1	Benign peripheral nerve sheath tumour	2.3%	0	1	9 years
1	Malignant peripheral nerve sheath tumour	2.3%	0	1	6 years
1	Haemangioma	2.3%	0	1	1.5 years
1	Compound melanocytoma	2.3%	0	1	4.5 years
Ferrets					
7	Mast cell tumour	33.3%	2	4	3 years
4	Squamous cell carcinoma	19%	0	4	3.8 years
4	Sebaceous epithelioma	19%	2	2	6.8 years
2	Poorly differentiated sarcoma	14.3%	1	1	6 years
1	Sebaceous carcinoma	4.8%	0	1	5 years
1	Secretory apocrine adenoma	4.8%	0	1	5 years
1	Apocrine adenocarcinoma	4.8%	0	1	4 years
1	Malignant peripheral nerve sheath tumour	4.8%	0	1	unspecified
Hamsters					
5	Trichoepithelioma	29.4%	2	3	15.2 months
3	Fibrosarcoma	17.6%	1	2	18.7 months
2	Melanoma	11.8%	2	0	9.5 months
1	Poorly differentiated sarcoma	5.9%	1	0	21 months
1	Trichofolliculoma	5.9%	0	1	23 months
1	Sebaceous epithelioma	5.9%	0	1	13 months
1	Apocrine adenocarcinoma	5.9%	0	1	12 months
1	Apocrine ductal adenocarcinoma	5.9%	0	1	15 months
1	Simple lipoma	5.9%	1	0	12 months
1	Epitheliotropic T-cell lymphoma	5.9%	0	1	18 months
Degus					
2	Fibrosarcoma	25%	1	0	6 years
2	Poorly differentiated sarcoma	25%	0	2	4.3 years
1	Myxosarcoma	12.5%	0	1	5 years
1	Anaplastic sarcoma with giant cells	12.5%	1	0	1.5 years
1	Malignant peripheral nerve sheath tumour	12.5%	1	0	3.5 years
1	Epitheliotropic T-cell lymphoma	12.5%	1	0	5 years
African pygmy hedgehogs					
1	Fibrosarcoma	20%	1	0	4 years
1	Liposarcoma	20%	0	1	4 years
1	Malignant peripheral nerve sheath tumour	20%	0	1	2 years
1	Histiocytic sarcoma	20%	0	1	7 years
1	Poorly differentiated sarcoma	20%	1	0	5 years
Mongolian gerbils					
2	Scent gland epithelioma	66.7%	0	2	2 years
1	Histiocytic sarcoma	33.3%	1	0	unspecified
Chinchillas					
2	Fibrosarcoma	66.7%	1	1	15.5 years
1	Haemangioma	33.3%	0	1	unspecified

## Data Availability

The datasets supporting the conclusions of this article will be made available by the authors, without undue reservation.

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
