# Peer review of "Cutaneous and Subcutaneous Tumours of Small Pet Mammals—Retrospective Study of 256 Cases (2014–2021)"

_animals, 2022, doi:10.3390/ani12080965_

Round 1

Reviewer 1 Report

In this paper, the authors present their work about cutaneous and subcutaneous tumors of small pet mammals (more than 10 species). This is a retrospective study based on the histopathologic analysis of 256 cases. Epidemiological data (species, breed, age, sex) are given when available. There is no information regarding follow-up or treatment.

First, I would like to thank the authors for their manuscript: it is well written (british English) and easy to read, with good quality pictures. The aim of the study is clear and results are well presented and discussed with a nice review of the literature.

Despite these qualities, the main weakness of this study is the total number of cases considering the high number of species. As a consequence, the number of tumors for each species is often much lower than in previous series dedicated to a particular species. In this context, the authors’ conclusions, in particular regarding the possible evolution of tumor incidence over the years, have to be taken very cautiously. That being said, the authors also provide the first (but short and therefore incomplete) description of some tumor type in particular species. Of note, some tumors were diagnosed based only on HE sections whereas the definitive diagnosis may require immunohistochemistry as the morphology can be equivocal with other tumors (peripheral nerve sheath tumors for example).

As a summary, although not original in its design and main results, the paper is well written and presented, and provides some useful additions to the literature.

Following are more detailed comments regarding the manuscript:

Lines 14-15: I would rather say that “Malignant tumors represented the predominant group of cutaneous tumours in rats……”.

Line 18: “Integumental histiocytic sarcoma” sounds odd. I would rather say “cutaneous histiocytic sarcoma”.

Line 20-21: Why “worrisome”? I don’t think this comment is useful. If the authors want to highlight the importance of their results, I would rather say that the high incidence of malignant neoplasms suggested by their results should incite owners and vets not to neglect cutaneous masses in these species.

Lien 38-39: do not account for a significant percentage in all species.

Line 42: would add “tumor” as a keyword

Lines 85-95: This is usually better presented in a table.

Line 151: Is “schwannoma” an appropriate term in rabbits? Has it been used in the literature. There is a lot of debate regarding the use of “schwannoma” in animals and “benign peripheral nerve sheath tumor” is usually preferred.

Line 205: I would have been very interested in a histological figure from the epithelioid haemangioma as this is an uncommon tumor in general and the first description in this species.

Line 369-370: I think it is unlikely to be the only explanation…

Line 387-390: At no time the authors demonstrate an increase in incidence. The number of cases is too low to conclude and the comparison with previous studies (different countries, different approaches etc.) warrants caution.

Reviewer 2 Report

The manuscript describes the skin pathology archive of a diagnostic laboratory in Poland. The information is useful, although I advise some changes.

One of the conclusions made in the study (eg line 20) is that a proportion of skin neoplasms in small mammals are malignant. It should be stated that this could be because small mammals get lots of malignant tumors. However, it could also be because this is a survey of histological submissions. Vets are much more likely to remove a skin mass if it looks malignant compared to a mass that looks benign. Therefore, generalizations about skin masses cannot be made. All the authors can conclude is that, of the skin masses submitted for histological evaluation, a proportion of these were malignant. This has to be made much clearer in the manuscript.

The authors should also make it clearer that this survey was only done at one location. There is no evidence that the results from Poland would be representative from other locations in the world. I’m not saying the results would be different, but it must be stressed that these results are only from a small number of pathologists in one diagnostic lab in one country.

Line 23. Remove the word ‘constantly’ and make it clear whether you are talking about Poland or globally.

Line 24. Did you see an increase in histology submissions between 2014 and 2021? I could not find the number of submissions/year so the statement saying that samples submitted for histology is increasing was not supported by the data presented.

Line 53. Many animal models exist for human diseases, but this review is of pet animals so this statement seems unnecessary and can be removed.

Line 56-65. The paragraph is hard to read as multiple species of pet small mammals are listed multiple times.

Line 66. Again, make it clear this is just for one location.

Line 123. Mammary gland fibroadenomas are by far the most common tumor of rats that appears as a lump in the skin. Presumably these were excluded in this study? If so, this should be stated as most people will know how frequently these develop in both female and male rats. Alternatively, if these were not excluded could it be that vets are so aware of these that they do not bother sending samples in for histology?

Line 300. The vast majority of feline cutaneous mast cell tumors are benign.

Line 387. The statement about malignant neoplasms increasing in small mammals isn’t supported by anything within the manuscript I could find. Is this between 2014 to 2021? If so, statistics should be done proving the rate of malignancies is increasing. If this is compared to other studies, this should be discussed in a way that the reader can judge if comparisons are valid and make it clear what percentage of neoplasms in each study was benign and malignant to support your statement that malignant neoplasms are getting more frequent. As before, you would then have to qualify this with the idea that vets are probably getting better at diagnosing benign neoplasms clinically (or doing cytology) which could explain why submissions of malignant neoplasms could be increasing to the diagnostic lab.

Reviewer 3 Report

Dear Authors,

I read carefully the manuscript intitled “Cutaneous and subcutaneous tumours of small pet mammals - 
retrospective study of 256 cases (2014-2021)”

I believe that the topic is very interesting, since in literature few data are available about tumors of small pet mammals. The work shows a large and interesting collection of cases of cutaneous and subcutaneous tumors in small mammals, and argues very well the differences with the pre-existing data in the literature species by species among those examined. The study is clear, the experimental design is well explained, the number of samples adequate, the results are well presented, and the pictures are of good quality. Moreover, the introduction successfully contextualizes the issue and the discussion provide sufficient argument on the topic and the results. The language is appropriate to my knowledge and the literature is recent and appropriate. I only have few minor comments that the authors should addressed before considering the manuscript suitable for publication in the review.

Below you can find some minor comments that you should address before considering the manuscript suitable for publication.

Minor comments:

L.105-L.124-L.143-L.158-L.172-L.191-L.196: insert range

LL.106-107 and LL.144-145 Data about breed were known in too few cases, I believe that this information is almost useless and weight down the text.

  1. 111-112: how about the other localisations?

L.107: insert total number of benign adnexal tumors in order to make the results more understandable

L.125: insert total number of soft tissue sarcomas

L.178: insert total number of mesenchymal tumors

Due to the paucity of information regarding cutaneous and subcutaneous tumors in chinchilla, I think it would be interesting to see a picture of tumors from them as well.
